# Automatic Segmentation of the Olfactory Bulb

**DOI:** 10.3390/brainsci11091141

**Published:** 2021-08-28

**Authors:** Dmitriy Desser, Francisca Assunção, Xiaoguang Yan, Victor Alves, Henrique M. Fernandes, Thomas Hummel

**Affiliations:** 1Smell & Taste Clinic, Department of Otorhinolaryngology, Technische Universität, 01307 Dresden, Germany; yanxguang01@outlook.com (X.Y.); thomas.hummel@tu-dresden.de (T.H.); 2Department of Informatics, School of Engineering, University of Minho, 4704-553 Braga, Portugal; francisca.s.assuncao@gmail.com (F.A.); valves@di.uminho.pt (V.A.); 3Center for Music in the Brain, Department of Clinical Medicine, Aarhus University, Noerrebrogade 44, 1A, 8000 Aarhus, Denmark; henrique.fernandes@clin.au.dk; 4Center of Functionally Integrative Neuroscience, Aarhus University, Noerrebrogade 44, 1A, 8000 Aarhus, Denmark; 5Flavour Institute, Department of Clinical Medicine, Aarhus University, Noerrebrogade 44, 1A, 8000 Aarhus, Denmark

**Keywords:** olfactory bulb, olfactory loss, deep learning, segmentation

## Abstract

The olfactory bulb (OB) has an essential role in the human olfactory pathway. A change in olfactory function is associated with a change of OB volume. It has been shown to predict the prognosis of olfactory loss and its volume is a biomarker for various neurodegenerative diseases, such as Alzheimer’s disease. Thus far, obtaining an OB volume for research purposes has been performed by manual segmentation alone; a very time-consuming and highly rater-biased process. As such, this process dramatically reduces the ability to produce fair and reliable comparisons between studies, as well as the processing of large datasets. Our study aims to solve this by proposing a novel methodological framework for the unbiased measurement of OB volume. In this paper, we present a fully automated tool that successfully performs such a task, accurately and quickly. In order to develop a stable and versatile algorithm and to train the neural network, we used four datasets consisting of whole-brain T1 and high-resolution T2 MRI scans, as well as the corresponding clinical information of the subject’s smelling ability. One dataset contained data of patients suffering from anosmia or hyposmia (N = 79), and the other three datasets contained data of healthy controls (N = 91). First, the manual segmentation labels of the OBs were created by two experienced raters, independently and blinded. The algorithm consisted of the following four different steps: (1) multimodal data co-registration of whole-brain T1 images and T2 images, (2) template-based localization of OBs, (3) bounding box construction, and lastly, (4) segmentation of the OB using a 3D-U-Net. The results from the automated segmentation algorithm were tested on previously unseen data, achieving a mean dice coefficient (DC) of 0.77 ± 0.05, which is remarkably convergent with the inter-rater DC of 0.79 ± 0.08 estimated for the same cohort. Additionally, the symmetric surface distance (ASSD) was 0.43 ± 0.10. Furthermore, the segmentations produced using our algorithm were manually rated by an independent blinded rater and have reached an equivalent rating score of 5.95 ± 0.87 compared to a rating score of 6.23 ± 0.87 for the first rater’s segmentation and 5.92 ± 0.81 for the second rater’s manual segmentation. Taken together, these results support the success of our tool in producing automatic fast (3–5 min per subject) and reliable segmentations of the OB, with virtually matching accuracy with the current gold standard technique for OB segmentation. In conclusion, we present a newly developed ready-to-use tool that can perform the segmentation of OBs based on multimodal data consisting of T1 whole-brain images and T2 coronal high-resolution images. The accuracy of the segmentations predicted by the algorithm matches the manual segmentations made by two well-experienced raters. This method holds potential for immediate implementation in clinical practice. Furthermore, its ability to perform quick and accurate processing of large datasets may provide a valuable contribution to advancing our knowledge of the olfactory system, in health and disease. Specifically, our framework may integrate the use of olfactory bulb volume (OBV) measurements for the diagnosis and treatment of olfactory loss and improve the prognosis and treatment options of olfactory dysfunctions.

## 1. Introduction

As one of the five basic human senses, the sense of smell plays an important role in our daily life. It is essential for the detection of dangers, such as gas, fire, smoke, or hazardous chemicals, and the quality of our everyday social life. Moreover, smelling odors makes up a major part of our pleasant experiences, whether eating a favorite meal, walking outside smelling blooming flowers and trees, or during intimacy with one’s partner. Last but not least, the sense of smell plays a fundamental role in some professions such as chef (approximately 0.5% of the German workforce), baker, or perfumer [1].

Clinical reviews have shown that 3–20% of the general population are affected by anosmia (complete loss of sense of smell) or hyposmia (reduced sense of smell) [2,3]. Olfactory deficits are associated with numerous neurodegenerative disorders such as Parkinson’s disease or Alzheimer’s disease and appear as prodromal symptoms. Moreover, patients with complete or partial olfactory loss have a higher risk of exhibiting symptoms of depression. Inversely, depressive symptoms are correlated with lower activation in structures involved in the olfaction perception pathway or with a decreased olfactory bulb (OB) volume [4]. Therefore, there is a need to understand the functioning of the human olfactory system.

Olfactory perception begins as the volatile odor molecules inhaled from the air bind olfactory receptor proteins in the cilia of olfactory sensory neurons housed in the neuroepithelium of the nasal cavity. This neuroepithelium contains 6–10 million neurons [5]. The axons of these neuroepithelium cells ascend through the cribriform plate as fila olfactoria to the OB located in the olfactory fossa of the ethmoid bone. The OB is the first stage of the olfactory signal processing system and, therefore, an essential part of the olfactory pathway. Signals from activated neuroepithelium cells are transmitted to the OB and then to primary olfactory regions of the brain such as the piriform cortex, entorhinal cortex, and amygdala. The outputs from the primary olfactory areas are then sent to other brain structures such as the orbitofrontal cortex (OFC), insula, and hippocampus [6].

It has been shown that OB volume correlates with olfactory sensitivity and it is decreased in patients with olfactory disorders [7]. Moreover, previous studies have shown that the OB volume correlates with the volume and grey matter density of the primary olfactory region [8]. From the range of known causes of primary olfactory loss, patients with post-traumatic and post-infectious anosmia or hyposmia consistently present a reduced OB volume when compared to healthy individuals [9]. About two thirds of the patients with congenital anosmia do not have a detectable OB on magnetic resonance imaging (MRI) scans, and one third present hypoplastic OB [10,11]. It has been shown that the decrease in OB volume correlates with decreased olfactory sensitivity. Reverse to this, an increase in OB volume, e.g., after endoscopic nasal surgery, leads to increased olfactory sensitivity [12]. All these findings suggest that the volume of the OB is an important marker for olfactory function and a predicting factor for the treatment of olfactory disorders [13].

The OB is a very small structure within the human brain with a volume ranging between 35–100 mm^3^ in normosmic individuals [13,14]. Therefore, to assess the volume of such a small structure, specialized high-resolution MR sequences are needed. However, the location of the OB within the olfactory fossa of the ethmoid bone makes it vulnerable to susceptibility artifacts. Susceptibility artifacts are distortions due to local magnetic field inhomogeneities and often arise at interfaces of tissues with different proton densities. Especially in fast sequences with lower spatial resolution, it is extremely challenging to achieve a satisfactory level of image quality necessary for performing image segmentation of small anatomical structures and estimating its volume. In clinical practice and research, the T2 coronal sequence is most commonly used because of its contrast between the OB’s tissue and the cerebrospinal fluid (CSF) in the surrounding area and because it is less vulnerable to susceptibility artifacts. Currently, the gold standard strategy for the measurement of OB volumes involves the manual segmentation of the visible OB in the coronal plane view slice by slice. Different segmentation techniques are known, such as manually tracing the outlines of the bulbs or highlighting the entire visible area slice by slice [15]. Still, manual segmentation is an extremely time-consuming process exhibiting variations in the degree of inter-observer and intra-observer reliability. It takes a well-experienced rater about 10–15 min to complete the OB segmentation of a single subject.

Over the last few years, machine learning (ML) algorithms have provided efficient solutions for automatic image segmentation. These algorithms have the potential to efficiently process more data and increase the reliability and repeatability of the results. Here, we applied ML to neuroimaging data to produce one of the first models that can automatically and accurately segment the OB, a very small anatomical region in the human brain [16,17].

In short, our algorithm starts by performing multimodal data preprocessing to localize the OB and compute the bounding box. Subsequently, a 3D U-Net model segments the OB within the bounding box. The performance of the algorithm was evaluated using established metrics [18]. Additionally, an experienced independent and blinded rater rated the quality of the agreement between the segmentations produced by both the algorithm and the human raters. To prove that the algorithm performs well independently of the field of view (FOV), image orientation, angulation, or other acquisition parameters, we tested our model on randomly selected subjects from neuroimaging datasets from previous studies.

## 2. Methods

### 2.1. Study Population

In this study, we used data from four different datasets [19,20,21]. All data were obtained at the University Hospital Carl-Gustav Carus in Dresden, Germany. Within the context of the respective studies, all participants signed informed consent on the use of their data for research purposes. All of these studies had been approved by the Ethics Committee at the University Clinic of the TU Dresden.

Data were separated into the following two sub-datasets: anosmia and healthy controls. The first dataset was collected on 79 patients between August 2015 and July 2017. Patients were diagnosed with anosmia or hyposmia. Healthy controls’ (*n* = 91) sub-dataset contained data from three different MRI studies.

In both datasets, participants’ smell ability was measured psychophysically using the Sniffin’ Sticks test [22]. The scores of the test (range: 1–48) were used to categorize participants’ smelling ability in the following categories: functional anosmia (TDI ≤ 16), hyposmia (16 > TDI < 30.75), or normosmia (TDI ≥ 30.75) [23] (Table 1).

### 2.2. MRI Image Acquisition

All MRI scans were acquired on a 3T Siemens Verio scanner (Siemens, Erlangen, Germany). All acquisitions were made using a 32-channel head coil. The following two modalities were used in this study: T1-weighted axial whole-brain scans and T2-weighted high-resolution coronal scans for OB imaging.

T1-weighted MPRAGE sequence was acquired using the following parameters: repetition time: 2300 ms; echo time 2.98: ms; flip angle: 9°; field of view: 240 × 256; acquisition matrix: 250 × 256 voxel size: 1 × 1 × 1 mm^3^; slice thickness: 1 mm; slices: 176. T2-weighted high-resolution sequence was acquired using following sequence parameters: repetition time: 5500 ms; echo time: 110 ms; flip angle: 150°, field of view: 120 × 120; acquisition matrix: 256 × 256; voxel size: 0.47 × 0.47 × 1.2 mm^3^; slice thickness: 1.2 mm (no gap).

### 2.3. Manual Segmentations of OB Volume

For both datasets, manual segmentation of the OBs was performed on T2-weighted images in coronal plane view using ITK-SNAP Software v. 3.6 [24]. First, both T1-weighted and T2-weighted scans were converted from DICOM format to g-zipped NIFTI format (nii.gz) using the dcm2niix conversion tool [25]. All measurements were performed by two raters independently. Both raters used the protocol for performing manual segmentations. Voxels belonging to the left and right OB were labeled with values 1 and 2, respectively. The manual measurements were saved as binary segmentation masks in g-zipped NIFTI format (nii.gz). To access the information for each label, respectively, the binary segmentation masks were converted to a NumPy array with nibabel python library [26,27]. The volumes of the labels for left and right OBs were calculated by multiplying the number of voxels of the label by voxel dimensions using the NumPy python library [28]. For all manual segmentation masks, the DC was calculated using the MedPy python library [29] to estimate the level of overlap between the two raters.

### 2.4. Automated Localization of the OBs

OBs have very small volumes, especially compared to the entire scanned brain volume. This leads to highly imbalanced data due to class imbalance between voxels labeled as foreground (voxels with values one/two for left/right OB) and voxels labeled as background (zero value voxels for any other tissue). To solve this issue, we opted for using a template-based approach. To the best of our knowledge, no earlier studies have normalized individual OBs to a standard stereotactic Montreal Neurological Institute (MNI) [30,31] space. Here, we developed a pipeline that allows the automatic transformation of all manual segmentations from T2 native space to MNI space. First, the T1 whole-brain image was co-registered to ICBM 2009c Nonlinear Asymmetric T1 MNI template image using ANTs SyN nonlinear registration tool [32]. Subsequently, the T2 image was co-registered to the T1 image by ANTs affine registration function. For both steps, the transformation matrices were saved and applied to the manual segmentations in native T2 space. Applying the inverted registration matrices allowed a two-step transformation of manual segmentation masks from T2 native space to MNI space. This procedure was extended to the manual segmentations produced by the two raters. The resulting binary masks in MNI space were transformed to NumPy arrays using nibabel. The NumPy arrays were added to one NumPy array and divided by the highest value in the array.

This process resulted in a probability map for the OBs in MNI space. To calculate the coordinates of the center of gravity (COG) of the OB, we applied a threshold of 0.5 to the resulting OB probability map and calculated the coordinates in MNI space (xyz: −4/44/−36) using the SciPy python library [33], and saved the result as a binary image (Figure 1).

### 2.5. Preprocessing Pipeline

The preprocessing pipeline of the segmentation tool performs all necessary steps to create normalized data for 3D U-Net input. As described above, the first step consists in co-registering the T1 image to MNI template image and the T2 image to T1 image to obtain the resulting transformation matrices. These will subsequently be inverted, and the resulting inverted transformation matrices applied to the COG binary map resulting in the COG in T2 native space. To reduce the number of features (voxels) and the data imbalance of the input images for processing in 3D U-Net, we constructed a bounding box extraction algorithm based on the COG in T2 native space. To ensure that all images have the same orientation in three-tridimensional space, all T2 images and the COG binary mask in T2 native space were reoriented using nibabel python library to canonical orientation. The bounding box edges were defined as points in 3D space shifted from the COG by +/−10 mm in the x-direction, +/−15 mm in the y-direction, and +/−5 mm in the z-direction. Afterwards, the array inside the calculated bounding box was extracted from the T2 image and the corresponding manual segmentation. The resulting images were resampled to a common voxel dimension of (0.5, 1, 0.5) and image shape of (4, 32, 32) using cubic interpolation function from fslpy python library (Figure 2) [34].

### 2.6. Training of the 3D U-Net Model

The entire model training process was carried out using the Monai Python Library [35]. Model training was performed on the dataset resulting from preprocessing pipeline described above. The dataset included the preprocessed data from 159 T2 images and 318 manual segmentations (159 manually created binary masks by two raters). Seven subjects from the anosmia dataset were excluded, due to not having a visible OB. Therefore, the manual segmentations made by both raters were empty, containing only the background. First, the dataset was randomly split into a training dataset (*n* = 191 subjects; 60% of the dataset), a validation dataset (*n* = 64 subjects; 20% of the dataset,), and a test dataset (*n* = 64 subjects; 20% of the dataset). First, all data underwent intensity normalization, a standard step of the monai preprocessing pipeline. To perform data augmentation, a random affine transformation was applied to the normalized data from the training dataset only. These transformations were automatically applied by RandAffined function from the monai python package during each epoch of the model training process. Specifically, the transformation features included a translation range of −10 to +10 voxels in each direction and rotation of −30° to +30° degrees using bilinear interpolation function for T2 images and nearest-neighbor interpolation for binary label masks.

The U-Net model was imported from monai python library and specified using following parameters: dimensions: 3; input channels: 1; output channels: 2; channels: 16, 32, 64, 128, 256; strides: 2, 2, 2, 2; num of residual units = 2.

The model was trained over 300 epochs. Each epoch contained 86 iterations of mini batches containing two corresponding image and binary mask pairs of size 64 × 32 × 32 voxels. We opted for the Tversky function to measure loss aversion. Network weights were optimized using the Adam optimization function (Ir = 0.001). One input channel and two output channels were defined for the input–output data stream of the model, containing three dimensions (Figure 3).

### 2.7. Postprocessing Pipeline

We designed and implemented a novel postprocessing pipeline to perform data cleanup of the primary output of the trained 3D U-Net model and transform the output to the original T2 image’s size, resolution, and orientation. As a first step, all clusters are relabeled to unique values. Secondly, clusters are thresholded based on their size to eliminate spurious clusters. Finally, the resulting mask is resampled to the resolution, voxel size, and orientation of the corresponding T2 image. This post-processed binary mask is then saved into an empty NumPy array of the same size as the T2 image—the final binary segmentation mask.

### 2.8. Evaluation of Training and Testing

The performance of the automatic segmentation was evaluated on the testing dataset. This dataset was kept separate before the model training step and, therefore, contained only data “unseen” by the model, consisting of 64 subjects and the corresponding manual segmentations of two raters. Performance was evaluated by computing an extensive set of statistical metrics—the dice coefficient (DC) and average symmetric surface distance (ASSD)—to evaluate the similarity between the model’s predicted segmentations and the manual segmentations, for both left and right OB.

DC is an overlay similarity index that reflects size and localization agreement and ranges from 0 (no overlap) to 1 (complete overlap). ASSD represents the mean distance of the binary objects in two images. Therefore, an ASSD of 0 mm represents a perfect match between both segmentations.

For subjects with diagnosed congenital anosmia and absence of the OBs, it was not possible to compute the DC because empty segmentation masks contain only the label corresponding to the image background.

Furthermore, to ensure the quality of predicted segmentations, all data were randomized and anonymized to ensure unbiased rating of the manual and predicted segmentations. A well-experienced rater, who had not been involved in any stage of the data segmentation process, was asked to rate the overlap of both manual segmentations and predicted masks using a performance scale. The rating criteria were defined as a scale: 1: “No congruency”, 2–3: “Poor congruency”, 4–5: “good congruency”, 6–7: “very good congruency”, 8–10: “excellent congruency”.

## 3. Results

Firstly, we compared the volumes of the manual segmentations made by rater one and two and calculated the DC for the left and right OBs individually (Table 2). For the dataset containing the anosmia patients’ data, the average DC was 0.77 ± 0.07 and 0.74 ± 0.10 mm^3^ for the left and right OB, respectively. The mean volume of the left OB was 37.82 ± 11.48 and 47.82 ± 14.78 mm^3^ measured by raters one and two, respectively. For the right OB, the mean volume of the segmentations produced by raters one and two was 34.32 ± 10.93 and 46.47 ± 15.43 mm^3^, respectively.

For the dataset containing the healthy control subjects’ data, the average DC for the left OB was 0.81 ± 0.06 mm^3^ and 0.80 ± 0.05 mm^3^ for the right OB. The mean volume for the left OB was 44.14 ± 12.38 mm^3^ measured by rater one and 56.01 ± 16.92 mm^3^ measured by rater two. The mean volume for the right OB was 42.47 ± 13.54 mm^3^ measured by rater one and 54.15 ± 17.66 mm^3^ measured by rater two (Figure 4).

For the evaluation of the overlap between the predicted segmentations of the algorithm and the manual segmentations of the OB for the subjects in the test dataset (N = 64), various metrics were calculated for the left and right OBs, separately, as well as for the entire segmentation mask. As it can be found in Table 3, the mean DC was 0.77 ± 0.05 (left OB: 0.78 ± 0.06, right OB: 0.75 ± 0.08) and the mean symmetric surface distance (ASSD) was 0.43 ± 0.10 (left OB: 0.41 ± 0.10, right OB: 0.44 ± 0.14) (Table 3).

Moreover, we compared the volumes of the manual segmentations and the predicted segmentations generated by our algorithm, for the left and right OBs individually, for the test dataset. For the left hemisphere, the mean OB volume of the predicted mask was 44.80 ± 8.59 mm^3^ and 46.14 ± 12.70 mm^3^ for the manually segmented mask. For the right OB, the predicted OB volume was 46.73 ± 8.86 mm^3^ for the predicted mask and 42.63 ± 14.19 mm^3^ for the manual masks (Table 4).

Subsequently, an independent, unbiased, and well-experienced rater rated the manual segmentations produced by both raters, as well as the predicted segmentation of the algorithm (Figure 5) (Table 5). The mean score was 6.23 ± 0.87 and 5.92 ± 0.81 for rater one and two, respectively, and 5.95 ± 0.87 for the segmentations generated by our model.

Additionally, we tested the algorithm on subjects from the olfactory dysfunction dataset with diagnosed congenital anosmia (N = 7). As aforementioned, given that these subjects do not have a visible OB, the resulting manual segmentation masks were empty, containing only the background label. Our model was able to correctly identify all these extraordinary datasets as non-existent OB cases, with the generated masks only containing the background label, as expected.

In addition to the extended quality evaluation, we developed two interfaces for our algorithm. The tool is fully available via the pip python package management system, and the command-line interface is accessible in the terminal on UNIX-based operating systems (OS). For use in a Microsoft native OS, a Linux sub-system needs to be further installed. A visual interface is also available, allowing the user to select the paths to folders containing data comfortably at the distance of only a few mouse clicks.

## 4. Discussion

In this study, we have developed a ready-to-use solution for the automatic segmentation of human OBs using 3D U-Net. The algorithm localizes the COG of OBs, performs the segmentation, and calculates the volumes with nearly the same accuracy (DC = 0.77 ± 0.05) as two experienced raters independently (inter-rater DC = 0.79 ± 0.08).

Previous studies have shown a high inter-rater reliability using manual segmentation to measure OBV [36]. Although the convergence of volumetric measurements based on absolute volume can be high, the segmentation labels do not necessarily overlap. This fact makes the reproducibility of manual volumetric segmentations dependent on the observer and his expertise. Moreover, manual segmentation is a very time-consuming process. It takes an experienced rater about 15 min per subject. Our automated solution of OB segmentation takes about 3–5 min per subject, dependent on the system properties. Our study offers solid evidence that the OB can indeed be segmented automatically and quickly, matching the level of expertise of trained raters, while minimizing potential sources of bias. We have tested our algorithm on a dataset (N = 7) containing only patients with congenital anosmia and our model was able to accurately detect the absence of the OBs.

In this study, manual segmentations made by two independent raters were considered as ground truth. However, as it can be found in Figure 5, given the relatively low inter-rater DC of 0.79 ± 0.08, we can reasonably assume that there is an approximately 20% variation in the “ground truth” (gold standard method). Therefore, it is difficult to assess how precise each of these measurements are in representing the real volume of the OB. A potential reason for this may lie in the limited resolution of the MRI acquisitions, which are additionally biased by signal noise and artifacts. These may, therefore, constitute major methodological limitations with direct impact on the precision of the manual and automated measurements of the OB volume. This issue can only be addressed by developing better MRI hardware and improving the acquisition techniques. One of the main known limiting factors of the acquisition quality is the acquisition time, as longer acquisition times are needed to improve the spatial resolution and signal-to-noise ratio of the images.

Overall, our results showed that the segmentations produced by our algorithm for OB volume measurements could be used for performing clinical measurements both on healthy subjects and on different clinical populations suffering from various olfactory dysfunctions. To further validate our method, we aim to expand our dataset using additional MRI scans from upcoming studies at our department and other research institutions. This improvement would prospectively increase the accuracy of the model’s predictions. Hopefully, it would lead to establishing the described tool as a clinical standard in the diagnosis of olfactory diseases.

At the current stage of development and validation, our algorithm for measuring the OB volume may have immediate application to the clinical setup, and reliably offer support to the diagnosis and treatment of patients with olfactory dysfunction. The unbiased OB detection offers a faster, easier, and more reliable (less prone to rater-dependent/image-quality bias) strategy for producing morphological characterizations of the OB. Furthermore, it offers the opportunity to perform reliable follow-up studies, centered in the investigation of volumetric changes to the OB that may occur over time.

## 5. Conclusions

In this study, we developed a tool that can perform multimodal automatic segmentation of the OBs based on whole-brain T1 and T2 coronal high-resolution images. We demonstrated that our method provides reliable results, matching the accuracy of the gold-standard technique for OB segmentation (i.e., manual segmentations made by experienced raters). As such, our methodological framework could be directly implemented in clinical practice, as well as provide a crucial contribution to olfactory research by offering the opportunity to investigate large neuroimaging datasets of healthy and unhealthy populations.

This improvement may open up new important possibilities for characterizing and predicting the absolute OBV changes induced by treatment to olfactory disfunction, as well as provide new insights into important dynamic alterations occurring in the olfactory system, which is essential for advancing our fundamental knowledge of the olfactory system and enhancing the diagnosis and treatment of olfactory dysfunction. Furthermore, it may facilitate the integration of the use of OB volume measurements in the daily clinical routine practice for the diagnosis and treatment of olfactory loss, which may be essential to improve the prognosis and, in essence, treatment options.

## Figures and Tables

**Figure 1 brainsci-11-01141-f001:**
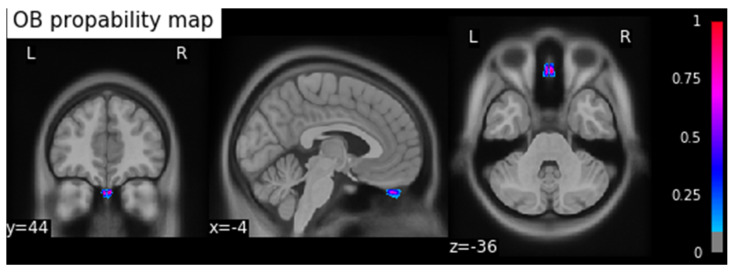
OB probability map: background image MNI ICBM 2009c Nonlinear Asymmetric template, threshold 0.5. The coordinates of the COG are xyz: −4/44/−36. (x: sagittal, y: frontal, z: axial orientations). The colormap indicates the probability from 0 to 1.

**Figure 2 brainsci-11-01141-f002:**
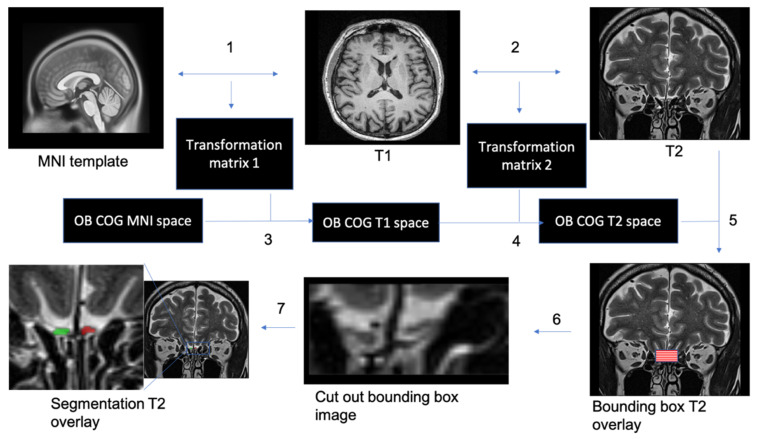
Automatic segmentation of the OBs. 1. T1 whole-brain image to MNI2009casym MNI template co-registration. 2. T2 high-resolution image to T1 whole-brain image co-registration. 3. Appling of inversed transformation matrix from step 1 to OB COG mask in MNI space. 4. Appling of inversed transformation matrix from step 2 to OB COG mask in T1 space. 5. Creating a 3D bounding box based on COG coordinates in T2 space from step 4. 6. Cut out the bounding box from the T2 image. 7. Performing image segmentation using trained 3D U-net.

**Figure 3 brainsci-11-01141-f003:**
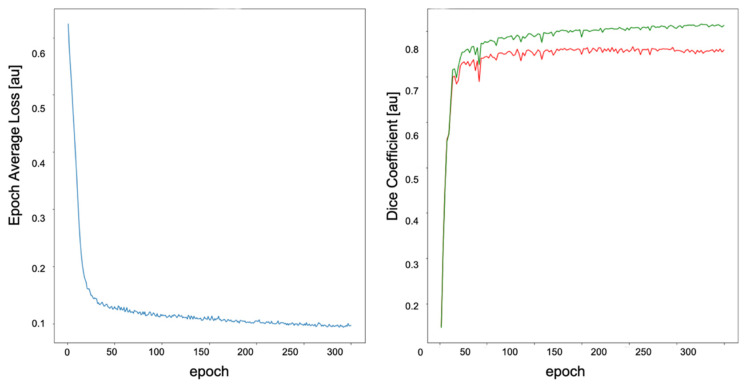
Model training process. **Left**: Training loss function. **Right**: DC plot for training (green curve) and validation (red curve) datasets. The DC and average loss values are both arbitrary units. The curves reach a plateau at epoch 100. The model at the epoch 297 was selected as the highest DC (0.84) of the training process. The unit (au) means arbitrary unit.

**Figure 4 brainsci-11-01141-f004:**
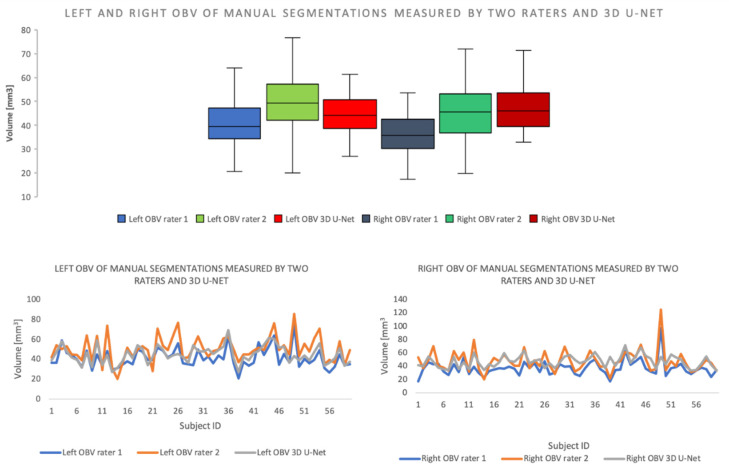
**Top**: Boxplot showing the OBVs of manual segmentations measured by two independent raters and the 3D U-Net model on test dataset. **Bottom left** and **right**: XY-Plot of the left and right OBV measured by two independent raters and 3D U-Net on test dataset. The trend of the graphs shows a high level of congruency of the measurements.

**Figure 5 brainsci-11-01141-f005:**
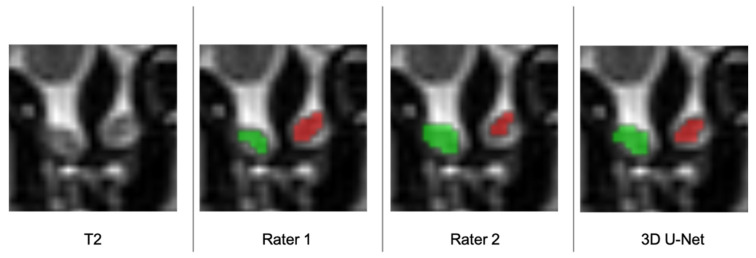
Automatic segmentation of the left (red) and right (green) OB in coronal T2 MRI scan performed manually by two independent raters and the algorithm.

**Table 1 brainsci-11-01141-t001:** Participant’s TDI Scores (sum score (TDI) for threshold (T), discrimination (D), and identification (I) of the odors; means (M), standard deviations (SD).

**Smell Dysfunction Patients (N = 79)**				
	**TDI**	**T**	**D**	**I**
M	18.00	2.67	7.81	7.52
SD	6.72	2.50	3.07	3.01
**Normosmic Controls (N = 91)**				
	**TDI**	**T**	**D**	**I**
M	36.39	9.97	12.92	13.50
SD	2.10	2.24	1.65	1.27

**Table 2 brainsci-11-01141-t002:** The table shows convergence of manual segmentations performed by two raters, based on DC and left and right olfactory bulb volume (LOBV and ROBV) for each rater, respectively. The unit (au) means arbitrary unit.

**Smell Dysfunction Patients**						
	**Left OB DC (au)**	**Right OB DC (au)**	**LOBV Rater 1 (mm^3^)**	**LOBV Rater 2 (mm^3^)**	**ROBV Rater 1 (mm^3^)**	**ROBV Rater 2 (mm^3^)**
M	0.77	0.74	37.82	47.64	34.32	46.47
SD	0.07	0.10	11.48	14.78	10.93	15.43
**Healthy Controls**						
	**Left OB DC (au)**	**Right OB DC (au)**	**LOBV Rater 1 (mm^3^)**	**LOBV Rater 2 (mm^3^)**	**ROBV Rater 1 (mm^3^)**	**ROBV Rater 2 (mm^3^)**
M	0.81	0.80	44.14	56.01	42.47	54.15
SD	0.06	0.05	12.38	16.92	13.54	17.66

**Table 3 brainsci-11-01141-t003:** Model performance metrics for the test dataset. DC (DC), Average Symmetric Surface Distance (ASSD). The unit (au) means arbitrary unit.

	DC (au)			ASSD (au)		
	Left OB	Right OB	Mean	Left OB	Right OB	Mean
M	0.78	0.75	0.77	0.41	0.44	0.43
SD	0.06	0.08	0.05	0.10	0.14	0.10

**Table 4 brainsci-11-01141-t004:** OB volumes of manual segmentations and predicted segmentations of the algorithm for the test dataset.

	Manual Segmentation		3D U-Net Segmentations	
	Left OB (mm^3^)	Right OB (mm^3^)	Left OB (mm^3^)	Right OB (mm^3^)
M	46.14	42.63	44.80	46.73
SD	12.70	14.19	8.59	8.86

**Table 5 brainsci-11-01141-t005:** Human validation scale. This table shows the manual rating criteria used by blinded rater for human validation of the segmentations. The rating scale was defined as: 1: “No congruency”, 2–3: “Poor congruency”, 4–5: “good congruency”, 6–7: “very good congruency”, 8–10: “excellent congruency.” The unit (au) means arbitrary unit.

Human Rating Results			
	Rater 1 (au)	Rater 2 (au)	U-Net (au)
M	6.23	5.92	5.95
SD	0.87	0.81	0.87
Minimum	4.00	4.00	4.00
Maximum	8.00	7.00	7.00

## Data Availability

The described Software is publically available at: https://doi.org/10.5281/zenodo.5283132 (accessed on 5 July 2021).

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
