# Peer review of "Automatic Segmentation of the Olfactory Bulb"

_brainsci, 2021, doi:10.3390/brainsci11091141_

Round 1

Reviewer 1 Report

This is a solid, well-scoped, well-organized, and well-written paper. It describes a method for automated segmentation of the olfactory bulb, in the category of applied machine learning. It is very appropriate for publication in Brain Sciences. In my opnion, no major revisions are indicated, though some minor edits are necessary.

In the comments below, the most important (and necessary) points to address are marked with (**). 

27: Does "remaining" refer to the other datasets, or the remaining patients in the first dataset? ((line 133 shows it's the first, but unclear here)

29 - 31: unclear to include manual ground truth and Algorithm in the same paragraph. Say eg "training labels were generated by ..."

63: I'm surprised you don't include the hyposmia/anosmia symptom of COVID. This would seem to be a major point of relevance.

74: a schematic (but spatially somewhat accurate) figure would be handy to visualize this paragraph

96: maybe add a parenthetical definition of susceptibility artifact

97: grammar (missing "it is", or change structure of "to achieve...")

111: and repeatability

112: typo

116: OB's voxels? or OB

(**) After 109 - 113: What is the prior work in automated segmentation of OBs (or closely-related tasks), and how does your method differ?
For example, you say (112) that yours is the first automated method. Is it rather one of the first? 
Papers to assess, compare/contrast, and cite (if their work is solid): 
1. Estrada, Reuter, et al. "Automated Olfactory Bulb Segmentation Pipeline for T2-Weighted MRI" 2021
2. Nouthout, Isgum et al. "Automatic segmentation of the olfactory bulbs in MRI" Feb 2021 (pretty recent)
3. Others (I did not do a lit review).
Also, if there are credible other papers on this topic, their results should be reported in Results, to give context to your results. It is not necessary to be "better", nor is a ML conference-style "bake-off" comparison table necessary; but it is necessary to acknowledge other valid work, and place your work in the communal context.

130, 133 to bottom of page: some formatting glitches

Table 1: add N (number of cases) for the two groups

157: Maybe note the approximate number of voxels in the OB in this imaging setup.  = 100 to 300? mean +/- std dev = ?

Python is capitalized (see Python citation specs)

171: maybe give a quick definition of Dice coefficient (ditto for Jaccard, Haussdorf, ASSD)

176 ff: clever method

191: this sentence is unclear, esp "added to one numpy array": I believe it means that all transformed arrays were combined into one MNI-based array. BTW, it seems that to get a probability map you would divide by the number of combined arrays, not the highest value, since if the most common voxel was only included in 9 out of 10 cases, its probability should be 0.9, not 1.0. But maybe these had the same effect.

194: Why use 0.5 as a threshold? Why not a weighted center: x_center = sum over all voxels i {x_i * v_i} where v_i is the probability of the i'th voxel?

217: could give the actual values here (20, 30, 4) or similar.

220: incomplete sentence

221: does not agree with line 156

235 (and previous): Inconsistent capitalization of Python and library names. Ditto for Dice coefficient.

Section 2.6:
(**) Can you give architecture details of the U-Net used, and add an appropriate reference?

236:  add "the"

244: the -> a random

246: add "the"

Fig 3: right hand plot: x-axis labels do not line up. Also: epoch 297 looks like it has the highest training DC, but not the highest validation DC. Is this an illusion, or did you use training set DC to choose a model?. Formatting: This figure could be cropped vertically to reduce white space, with maybe a broken line for epochs 0 - 10.

269: "process" -> "transform" or similar.

270: should be past tense?

291: unclear: ", carefully produced by..." Does this mean "to ensure that the manual and predicted segmentations were rated in an unbiased manner by an expert third rater" (or similar)?

294: Was this score 1 - 10 based on calculation of graphical annotations by the rater, or was it a subjective scale "eyeballed" by the rater after looking at the pair of images?

298: "raters 1 and 2"

300: typo

Table 2: (i) coefficient is mis-spelled (maybe elsewhere also). (ii) bold font formatting error. (iii) All tables might be clearer if results were shown as mean +/- std dev, eg "0.77 +/- 0.07". (iv) what is the difference between columns 3-4 and 5-6? The headings are identical. 

(**) lines 305 - 330. All these tables and figures have many errors and are unclear/inaccurate. Please post a corrected version.

330: typo "."  Ditto 345

330 - 334: Move this to line 294? Or remove "subsequently" to avoid narrative repetition with 294.

336, 337: typo. "Figure" should not be present.

Figure 4: Bottom plot: Is there a better way to index the subjects on the x-axis, eg sort by increasing mean(rater 1 OBV, rater 2 OBV). The random ordering makes a noisy, non-informative plot (this seems to be a common problem in ML papers). Top: you could match the colors by rater, and slightly separate LOBV and ROBV plots, for greater clarity. Caption:  "the test dataset"

Fig 5: Is this figure referred to in the text before line 392, in particular before it shows up? It's a good figure. Because it is anecdotal, it might work better earlier in the section, to put a face the results.

356: "In addition to"

(**) 356 ff: Important: Is this code publicly-available (if so, what is the repository), or is it proprietary?

382: applicability ... to predict -> the predictive ability

384: remove "have", from -> by

388: typo extra ","

392: found -> illustrated; It is shown in the relevant table.

(**) 406 (also 427): Is this statement true? Often, permitting akin to FDA-approval is required for clinical use. Use in studies might not require such an approval process.

420: This does not avoid rater bias (since the algorithm has its own bias). It avoids intra-rater variability, and avoids inter-rater variability affecting longitudinal studies (since human raters come and go).

438: usage of "in essence" is irregular here.

388 - 390: Sentence has incorrect grammar

309 - 318: Missing figures.

References: Please double-check that all is in order. Eg [21], [26] are incomplete. 

Author Response

Thank you very much for your comments and suggestions!

Reviewer 1:

This is a solid, well-scoped, well-organized, and well-written paper. It describes a method for automated segmentation of the olfactory bulb, in the category of applied machine learning. It is very appropriate for publication in Brain Sciences. In my opnion, no major revisions are indicated, though some minor edits are necessary.

In the comments below, the most important (and necessary) points to address are marked with (**). 

27: Does "remaining" refer to the other datasets, or the remaining patients in the first dataset? ((line 133 shows it's the first, but unclear here)

Done

Remaining refers to the other datasets

29 - 31: unclear to include manual ground truth and Algorithm in the same paragraph. Say eg "training labels were generated by ..."

Done

63: I'm surprised you don't include the hyposmia/anosmia symptom of COVID. This would seem to be a major point of relevance.

Unfortunately, we have not had enough data to include as the model was trained at the beginning of the pandemic.

74: a schematic (but spatially somewhat accurate) figure would be handy to visualize this paragraph

The pathway is difficult to visualize schematically because the anatomical structures and their anatomical locations and shape are extremely relevant. I will try to find an image for visualization of the pathswas.

96: maybe add a parenthetical definition of susceptibility artifact

Done

97: grammar (missing "it is", or change structure of "to achieve...")

Done

111: and repeatability

Done

112: typo

Done

116: OB's voxels? or OB

Here I mean olfactory bulbs

(**) After 109 - 113: What is the prior work in automated segmentation of OBs (or closely-related tasks), and how does your method differ?
For example, you say (112) that yours is the first automated method. Is it rather one of the first? 
Papers to assess, compare/contrast, and cite (if their work is solid): 
1. Estrada, Reuter, et al. "Automated Olfactory Bulb Segmentation Pipeline for T2-Weighted MRI" 2021
2. Nouthout, Isgum et al. "Automatic segmentation of the olfactory bulbs in MRI" Feb 2021 (pretty recent)
3. Others (I did not do a lit review).
Also, if there are credible other papers on this topic, their results should be reported in Results, to give context to your results. It is not necessary to be "better", nor is a ML conference-style "bake-off" comparison table necessary; but it is necessary to acknowledge other valid work, and place your work in the communal context.

Done (included)

130, 133 to bottom of page: some formatting glitches

Done

Table 1: add N (number of cases) for the two groups

Done

157: Maybe note the approximate number of voxels in the OB in this imaging setup.  = 100 to 300? mean +/- std dev = ?

In my opinion, the number of voxels does not represent the reference of OB volume correctly, as it depends on image resolution, therefore the reference is given in mm3

Python is capitalized (see Python citation specs)

Done (included)

171: maybe give a quick definition of Dice coefficient (ditto for Jaccard, Haussdorf, ASSD)

Included definitions of DC and ASSD

Exluded Jaccard, Haussdorf values, as they are not providing that much additional information about model performance in this case.

176 ff: clever method

Thank you :)

191: this sentence is unclear, esp "added to one numpy array": I believe it means that all transformed arrays were combined into one MNI-based array. BTW, it seems that to get a probability map you would divide by the number of combined arrays, not the highest value, since if the most common voxel was only included in 9 out of 10 cases, its probability should be 0.9, not 1.0. But maybe these had the same effect.

Done

194: Why use 0.5 as a threshold? Why not a weighted center: x_center = sum over all voxels i {x_i * v_i} where v_i is the probability of the i'th voxel?

Well, actually this was the only idea, that I had.. but it seemed to have worked too..

217: could give the actual values here (20, 30, 4) or similar.

(xyz coordinates are added)

Done

220: incomplete sentence

Done

221: does not agree with line 156

Could you maybe explain the mismatch between the two lines?

235 (and previous): Inconsistent capitalization of Python and library names. Ditto for Dice coefficient.

Done

Section 2.6:
(**) Can you give architecture details of the U-Net used, and add an appropriate reference?

Done

236:  add "the"

Done

244: the -> a random

Done

246: add "the"

Done

Fig 3: right hand plot: x-axis labels do not line up. Also: epoch 297 looks like it has the highest training DC, but not the highest validation DC. Is this an illusion, or did you use training set DC to choose a model?. Formatting: This figure could be cropped vertically to reduce white space, with maybe a broken line for epochs 0 - 10.

The plot was created directly after the train algorithm. Unfortunately, I am not able to “re-plot” it. At epoch 297 the overfitting seemed to be to high..

269: "process" -> "transform" or similar.

Done

270: should be past tense?

I am not a native speaker, but I was meaning passive voice here..

291: unclear: ", carefully produced by..." Does this mean "to ensure that the manual and predicted segmentations were rated in an unbiased manner by an expert third rater" (or similar)?

Done

294: Was this score 1 - 10 based on calculation of graphical annotations by the rater, or was it a subjective scale "eyeballed" by the rater after looking at the pair of images?

Eyeballed by the rater after looking at the 3 versions (rater 1, rater 2, unet) of the segmentations

298: "raters 1 and 2"

Done

300: typo

Done

Table 2: (i) coefficient is mis-spelled (maybe elsewhere also). (ii) bold font formatting error. (iii) All tables might be clearer if results were shown as mean +/- std dev, eg "0.77 +/- 0.07". (iv) what is the difference between columns 3-4 and 5-6? The headings are identical. 

Done

(**) lines 305 - 330. All these tables and figures have many errors and are unclear/inaccurate. Please post a corrected version.

Done

330: typo "."  Ditto 345

Done

330 - 334: Move this to line 294? Or remove "subsequently" to avoid narrative repetition with 294.

Done

336, 337: typo. "Figure" should not be present.

I am sorry, but I have not found the typo..

Figure 4: Bottom plot: Is there a better way to index the subjects on the x-axis, eg sort by increasing mean(rater 1 OBV, rater 2 OBV). The random ordering makes a noisy, non-informative plot (this seems to be a common problem in ML papers). Top: you could match the colors by rater, and slightly separate LOBV and ROBV plots, for greater clarity. Caption:  "the test dataset"

If I understood the comment correctly, the colors are already matched by rater or unet, aren’t they? On the x axis there are subject-ids.. like subject 1, 2, 3 and so on..

Fig 5: Is this figure referred to in the text before line 392, in particular before it shows up? It's a good figure. Because it is anecdotal, it might work better earlier in the section, to put a face the results.

Yes, but it shows the results.. therefore I have placed it into result section.
356: "In addition to"

Done

(**) 356 ff: Important: Is this code publicly-available (if so, what is the repository), or is it proprietary?

The code will of course published, I will add a link into the final version, because I need to discuss it with other co-authors. I think this process will take about 1 week.

382: applicability ... to predict -> the predictive ability

Done

384: remove "have", from -> by

Done

388: typo extra ","

Done

392: found -> illustrated; It is shown in the relevant table.

Done

(**) 406 (also 427): Is this statement true? Often, permitting akin to FDA-approval is required for clinical use. Use in studies might not require such an approval process.

Well, for clinical use it will need to be approved. The tool can already be applied for research porpose.

420: This does not avoid rater bias (since the algorithm has its own bias). It avoids intra-rater variability, and avoids inter-rater variability affecting longitudinal studies (since human raters come and go).

True. Changed.

438: usage of "in essence" is irregular here.

Done

388 - 390: Sentence has incorrect grammar

Done

309 - 318: Missing figures.

I am sorry, I have not understood what figures do you mean?

References: Please double-check that all is in order. Eg [21], [26] are incomplete. 

I have re-checked the references. [21], [26] It is suggested by the authors to cite zenodo publication of the code. The references were automatically created by Zotero. Could you maybe suggest what information should be added to the reference, please?

Reviewer 2 Report

This study compares manual with automated measurements of olfactory bulb volume. The authors applied machine learning algorithms to neuroimaging data to produce the first model which can automatically and accurately segment the olfactory bulb. The authors developed this tool to perform multimodal automatic segmentation of the olfactory bulbs based on whole-brain T1 and T2 coronal high-resolution images. The study provides evidence that the olfactory bulb can indeed be segmented automatically and quickly and matches the level of expertise of trained raters while minimizing potential sources of bias. The model was able to accurately detect the absence of olfactory bulbs when tested on patients with congenital anosmia.  The study contains relevant findings, and the model can serve as an important clinical tool since the study of the olfactory bulb is relevant for several neurodegenerative diseases because olfactory dysfunction can be a prodromal symptom.

At several places in the manuscript, formatting issues detract from the content of the study as indicated below.

Abstract, line 26: change ‘One dataset contained data of patients suffering from anosmia or hyposmia (N=79), and the remaining corresponding to cohorts of healthy controls (N=91).’  to ‘One dataset contained data of patients suffering from anosmia or hyposmia (N=79), and another dataset corresponds to cohorts of healthy controls (N=91).’  

Abstract, line 49, last sentence: OBV probably stands for OB volume but the abbreviation needs to be spelled out at its first use.

Introduction, line 71: change ‘Olfactory perception begins as the volatile odor molecules inhaled from the air bind on the neuroepithelium of the nasal cavity.’ To ‘Olfactory perception begins as the volatile odor molecules inhaled from the air bind to olfactory receptor proteins in the cilia of olfactory sensory neurons housed in the neuroepithelium of the nasal cavity.’

Line 73: change ‘This neuroepithelium is made of 6–10 million neurons [5]. The axons of this neuroepithelium cells…’ to ‘This neuroepithelium contains 6–10 million olfactory sensory neurons [5]. The axons of these neurons…’

Line 89: change ‘volme’ to ‘volume’

Line 130: what does the ‘1’ refer to?

Line 133: sentence is incomplete.  Formatting issue?

Below line 135: references appear in the text.  Formatting issue? Are these references to the datasets used?

Line 181: spell out MNI at its first use

Figure 1: I realize that the OB is small, but it is difficult to detect changes in the color map.  Is it possible to make the image larger or focus primarily on the OB?

Line 221: the brackets for references and voxel dimension/image shape are the same but should be different to distinguish one from the other.

Line 253: more method details would be helpful, e.g., what is the ‘Tversky function’?

Line 283: it would be helpful to have more of an explanation of the different statistical metrics, e.g., Haussdorf distance, ASSD, etc.

Line 310: what and where is figure 0?

Line 314: what and where is figure 64?

Line 330: change ‘Subsequently. an independent’ to ‘Subsequently, an independent’

Author Response

Thank you very much for your comments and suggestions!

This study compares manual with automated measurements of olfactory bulb volume. The authors applied machine learning algorithms to neuroimaging data to produce the first model which can automatically and accurately segment the olfactory bulb. The authors developed this tool to perform multimodal automatic segmentation of the olfactory bulbs based on whole-brain T1 and T2 coronal high-resolution images. The study provides evidence that the olfactory bulb can indeed be segmented automatically and quickly and matches the level of expertise of trained raters while minimizing potential sources of bias. The model was able to accurately detect the absence of olfactory bulbs when tested on patients with congenital anosmia.  The study contains relevant findings, and the model can serve as an important clinical tool since the study of the olfactory bulb is relevant for several neurodegenerative diseases because olfactory dysfunction can be a prodromal symptom.

At several places in the manuscript, formatting issues detract from the content of the study as indicated below.

Abstract, line 26: change ‘One dataset contained data of patients suffering from anosmia or hyposmia (N=79), and the remaining corresponding to cohorts of healthy controls (N=91).’  to ‘One dataset contained data of patients suffering from anosmia or hyposmia (N=79), and another dataset corresponds to cohorts of healthy controls (N=91).’  

Done

Abstract, line 49, last sentence: OBV probably stands for OB volume but the abbreviation needs to be spelled out at its first use.

Done

Introduction, line 71: change ‘Olfactory perception begins as the volatile odor molecules inhaled from the air bind on the neuroepithelium of the nasal cavity.’ To ‘Olfactory perception begins as the volatile odor molecules inhaled from the air bind to olfactory receptor proteins in the cilia of olfactory sensory neurons housed in the neuroepithelium of the nasal cavity.’

Done

Line 73: change ‘This neuroepithelium is made of 6–10 million neurons [5]. The axons of this neuroepithelium cells…’ to ‘This neuroepithelium contains 6–10 million olfactory sensory neurons [5]. The axons of these neurons…’

Done

Line 89: change ‘volme’ to ‘volume’

Done

Line 130: what does the ‘1’ refer to?

Changed. Done.

Line 133: sentence is incomplete.  Formatting issue?

Changed. Done.

Below line 135: references appear in the text.  Formatting issue? Are these references to the datasets used?

Changed. Done.

Line 181: spell out MNI at its first use

Done

Figure 1: I realize that the OB is small, but it is difficult to detect changes in the color map.  Is it possible to make the image larger or focus primarily on the OB?

Unfortunately, the image resolution does not allow a sufficient image quality for zooming into OB. Because of t1 anatomical image as background image, it is not possible to export the plot as vector image.

Line 221: the brackets for references and voxel dimension/image shape are the same but should be different to distinguish one from the other.

Done

Line 253: more method details would be helpful, e.g., what is the ‘Tversky function’?

There are different loss functions for model training optimization available in the monai python library. Unfortunately, I was not able to find a sufficient documentation and explanation for how this los function works in detail.

Line 283: it would be helpful to have more of an explanation of the different statistical metrics, e.g., Haussdorf distance, ASSD, etc.

Done

Line 310: what and where is figure 0?

Figure 0 does not exists. There are only figures 1-5.

Line 314: what and where is figure 64?

Figure 0 does not exists. There are only figures 1-5.

Line 330: change ‘Subsequently. an independent’ to ‘Subsequently, an independent’

Done